# The Xanthophyll Carotenoid Lutein Reduces the Invasive Potential of *Pseudomonas aeruginosa* and Increases Its Susceptibility to Tobramycin

**DOI:** 10.3390/ijms23137199

**Published:** 2022-06-28

**Authors:** Christian Emmanuel Mahavy, Adeline Mol, Blandine Andrianarisoa, Pierre Duez, Mondher El Jaziri, Marie Baucher, Tsiry Rasamiravaka

**Affiliations:** 1Laboratory of Biotechnology and Microbiology, University of Antananarivo, BP 906, Antananarivo 101, Madagascar; mitsinjomahavy@gmail.com (C.E.M.); blandine@andrianarisoa.net (B.A.); 2Laboratory of Plant Biotechnology, Université Libre de Bruxelles, 6041 Gosselies, Belgium; adeline.mol@ulb.be (A.M.); jaziri@ulb.ac.be (M.E.J.); marie.baucher@ulb.be (M.B.); 3Unit of Therapeutic Chemistry and Pharmacognosy, University of Mons, 7000 Mons, Belgium; pierre.duez@umons.ac.be

**Keywords:** all-*trans* lutein, antivirulence, biofilm, *Pseudomonas aeruginosa*, quorum sensing

## Abstract

Recently, the xanthophyll carotenoid lutein has been qualified as a potential quorum sensing (QS) and biofilm inhibitor against *Pseudomonas aeruginosa.* To address the potential of this xanthophyll compound as a relevant antivirulence agent, we investigated in depth its impact on the invasion capabilities and aggressiveness of *P. aeruginosa* PAO1, which rely on the bacterial ability to build and maintain protective barriers, use different types of motilities and release myriad virulence factors, leading to host cell and tissue damages. Our data, obtained on the PAO1 strain, indicate that all-*trans* lutein (Lut; 22 µM) disrupts biofilm formation and disorganizes established biofilm structure without affecting bacterial viability, while improving the bactericidal activity of tobramycin against biofilm-encapsulated PAO1 cells. Furthermore, this xanthophyll affects PAO1 twitching and swarming motilities while reducing the production of the extracellular virulence factors pyocyanin, elastase and rhamnolipids as well as the expression of the QS-regulated *lasB* and *rhlA* genes without inhibiting the QS-independent *aceA* gene. Interestingly, the expression of the QS regulators *rhlR*/*I* and *lasR*/*I* is significantly reduced as well as that of the global virulence factor regulator *vfr*, which is suggested to be a major target of Lut. Finally, an oxidative metabolite of Lut, 3′-dehydrolutein, induces a similar inhibition phenotype. Taken together, lutein-type compounds represent potential agents to control the invasive ability and antibiotic resistance of *P. aeruginosa*.

## 1. Introduction

Antibiotics are drugs of choice in the struggle against bacterial infectious diseases. However, their misuse and/or overuse induce an increased spread of multidrug resistance bacteria [1,2] which represent a worldwide threat to human health [3]. Bacterial resistance can be inherent (constitutive or induced), developed through mutation, and/or transmitted from strain to strain through encoding genetic material [4]; the reduction in antibiotic effectiveness may depend on distinct mechanisms, including the production of inactivating enzymes, the activation/induction of efflux pumps and the modification and/or alteration of a target site or outer membrane structure [5]. Beyond these common resistance mechanisms, bacteria have the capacity to develop into a biofilm, which is a multicellular community of bacteria adhering to each other and onto surfaces, and which is also included in an adhesive and protective matrix of extracellular polymeric substances (EPS) [6,7]. This bacterial behavior allows them not only to colonize biotic and abiotic surfaces, a crucial process to bacterial invasion, but also to resist against external aggression from biocides and antibiotics, representing a real obstacle in the fight against infections [8].

*Pseudomonas aeruginosa* is a Gram-negative bacterium that can easily form biofilm and exhibits a natural resistance to many conventional antibiotics [9,10]. This opportunistic human, animal and plant pathogen causes both acute and chronic infections by disseminating through its hosts thanks to different types of motilities, forming protective biofilms and producing myriad virulence factors. The regulation of *P. aeruginosa* invasive behaviors depends on a cell-to-cell communication system, termed quorum sensing (QS) [11], that allows bacteria to detect their critical cell density by producing and perceiving diffusible signal molecules (also called auto-inducers) in order to coordinate common responses [12]. *P. aeruginosa* possesses two important QS systems, *las* and *rhl*, based on acyl-homoserine lactones (AHLs) auto-inducers (3-oxo-C12-HSL and C4-HSL, respectively) [11]. Following their synthesis, that is catalyzed by acyl-homoserine lactone synthases (LasI and RhlI); these auto-inducers form transcription factor complexes with their corresponding receptors (LasR and RhlR), which upregulate the expression of QS-regulated genes implicated in biofilm development, motilities and the production of virulence factors, including pyocyanin, elastase and rhamnolipids, which leads to a strong expression of bacterial virulence [10,13]. The *las* and *rhl* systems are organized in a hierarchical manner such that the *las* system regulates the *rhl* system at the transcriptional and post-transcriptional levels [14,15,16]. Finally, the *las* and *rhl* systems are positively influenced by global regulators GacA (global antibiotics and cyanide control) and Vfr (virulence factor regulator) at both the transcriptional and post-transcriptional levels, so that disruption of these global regulators leads to a cascade of bacterial virulence inhibition [17,18,19,20]. With respect to the entanglement between biofilm development, resistance and virulence genes expression in *P. aeruginosa*, the search for compounds that target global virulence regulation could be valuable to extend the antimicrobial agents panel and strengthen the therapeutic arsenal against persistent *P. aeruginosa* infections.

For almost two decades, numerous natural and synthetic compounds have been identified and demonstrated to reduce *P. aeruginosa* virulence and biofilm formation. The mechanism by which these compounds interfere with, and reduce, bacterial virulence generally implies quorum quenching pathways. Indeed, virulence inhibitors act either (*i*) as antagonists/competitors of native AHLs, as is the case for AHL synthetic analogues (e.g., 3-oxo-C12-cyclohexanone) [21] and the flavonoids baicalein/baicalin [22,23], or (*ii*) as disruptors of QS regulator expressions at transcriptional and/or translational level; these notably include the semi-synthetic azithromycin, which reduces the expression of *gacA* [24], and the flavonoid quercetin, which reduces *vfr* expression [25], both leading to a cascade of inhibition in the QS circuitry and finally attenuating *P. aeruginosa* virulence. Additionally, as we previously reviewed, many natural extracts and compounds do interfere with bacterial virulence through quorum quenching pathways [10,26]. For instance, the flavanone naringenin at 4 mM and oleanolic aldehyde coumarate at 200 µM have been reported to inhibit both virulence factors production and biofilm formation in *P. aeruginosa* [27,28]. Likewise, terpenoids, and particularly the monocyclic diterpenoid cassipourol and the triterpenoid β-sitosterol, inhibit *P. aeruginosa* QS-regulated and QS-regulatory genes expression in *las* and *rhl* systems and disrupt the formation of biofilms, both at concentrations of 200 µM [29]. Recently, Sampathkumar et al. [30] reported that the xanthophyll carotenoid lutein at 35 µM inhibits biofilm formation (~60% inhibition), pyocyanin (~55% inhibition) and QS genes expression in *P. aeruginosa* PAO1. Moreover, the isomer of lutein, zeaxanthin (12 µM) also inhibits biofilm formation by ~40% and QS-regulated *lasB* and *rhlA* genes expression in PAO1 by ~60% [31]. Both studies highlight the anti-QS and antibiofilm potential of lutein and its isomer with an attractive low active concentration, stirring up our interest for their potential use as antivirulence agents. Consequently, the present study investigates in depth the potential of commercially available all-*trans* lutein (Figure 1) as an antivirulence agent and antibiotic adjuvant against PAO1, through their effect on biofilm development and maintenance, motilities, and virulence factors production, while defining their plausible non-bactericidal mechanism and QS target(s).

## 2. Results

### 2.1. All-Trans Lutein Reduces PAO1 Biofilm Formation in a Dose-Dependent Manner

Among antivirulence properties, the effect of all-*trans* lutein (Lut) on PAO1 biofilm formation was firstly assessed, according to its importance for bacterial colonization, invasion and resistance. Noticeably, there was a significant decrease, in a dose-dependent manner, of biofilm biomass when PAO1 wild-type was grown for 24 h in the presence of Lut. As shown in Figure 2, antibiofilm activity is greatly enhanced by increasing concentrations of tested compound (in the range of 1.38–88 µM) with an IC_50_ estimated at 12 µM. This result comforts the reported antibiofilm activity by Sampathkumar et al. [30] even if biofilm protocol assays were not similar, notably in growth medium composition (complex medium versus minimum medium in our study). To ensure that these antibiofilm effects are not due to a drop in bacterial viability, we evaluated the kinetic growth of PAO1 in the presence of Lut at 22 µM, concentration above the biofilm inhibition IC_50_ proportion (56 + 2% inhibition). As a result, the cell growth kinetics of PAO1 is not affected in comparison with the negative control (1% dimethylsulfoxide, DMSO) suggesting that bacterial development is not impaired at any stage of growth (Figure 3) in the presence of Lut; this concentration of 22 µM has then been applied for all subsequent experiments.

### 2.2. Lut Affects the Biofilm Phenotype of PAO1

Microscopic analysis allows deep visualization of the impact of Lut on the colonization ability of PAO1 onto surface. In epifluorescence microscopy, the biofilm architecture of PAO1 grown in static condition for 24 h with Lut appeared drastically disrupted compared to the control condition (DMSO, 1%). Actually, as shown in Figure 4, the control biofilm is composed of a thick and homogenous layer on coverslips with good and regular cell-to-cell connections interspaced by uncolonized surface (Figure 4A, DMSO), whereas biofilm formed in the presence of Lut (Figure 4A, Lut) exhibits a phenotype in which PAO1 fail to establish compact cell-to-cell attachment, resulting in a loss of microcolonies confluence and an important increase in uncolonized surface; this suggests a lack of biofilm biomass in coherence with the crystal violet staining measurements (Figure 2). As Lut clearly impairs the PAO1 biofilm formation, we further examined its effect on one-day-old pre-formed biofilms. As shown in Figure 4B, DMSO does not affect the pre-formed biofilm, whereas the addition of Lut leads to a drastic loss of the compact and heterogeneous biofilm layer, leaving a structure mainly composed of isolated bacterial clumps. Altogether, these results suggest that Lut affects not only biofilm formation but also its maintenance, leading to bacterial dispersion out of a pre-formed biofilm.

### 2.3. Lut Exhibits Antibiotic-Synergizing Activities in One-Day-Old PAO1 Biofilm

Since Lut affects both biofilm formation and structure maintenance, it is tempting to evaluate the protective ability of the Lut-perturbed biofilm matrix against environmental aggressors such as antibiotics; tobramycin has been selected for this test as it is widely used to treat acute *P. aeruginosa* exacerbations in patients with cystic fibrosis [32] but appears less effective in biofilm-encapsulated *P. aeruginosa* compared to their planktonic counterparts [33]. As shown in Figure 5, Lut in association with tobramycin significantly increased the effectiveness of the antibiotic against biofilm-encapsulated bacteria compared to the association tobramycin-DMSO. When Lut (22 µM) and tobramycin (107 µM, 50-fold above the planktonic MIC) were added to one-day-old biofilm cultures, the proportion of observed dead cells was 90 ± 3%, compared to 17 ± 4% for the association tobramycin-DMSO. These results suggest a serious improvement of antibiotic diffusion/penetration through the Lut-disrupted biofilm matrix.

### 2.4. Lut Affects PAO1 Swarming and Twitching Motilities

It is established that motilities are strongly associated with *P. aeruginosa* pathogenesis by contributing to colonization of different environments and attachment to surfaces [34]. As a means of bacterial movement, *P. aeruginosa* mainly uses two types of motilities, swarming and twitching, which also play an important role in the biofilm formation [12]. As shown in Figure 6, Lut significantly decreases swarming and twitching motilities (40 ± 9 and 34 ± 5% of inhibition, respectively) at 22 µM (Figure 6A,B), suggesting there is a difficulty for PAO1 to move onto humidified solid surface such as agar medium, mucous membranes or skin.

### 2.5. Lut Reduces Pyocyanin, Elastase and Rhamnolipids Production in PAO1

Invasive infection by *P. aeruginosa* is based not only on its ability to invade and disseminate through its hosts, thanks to motilities and biofilm establishment, but also to its ability to release myriad virulence factors such the phenazine pyocyanin, the metalloenzyme elastase and the biosurfactant rhamnolipids [10]. In this respect, Lut was investigated for its effect on pyocyanin, elastase and rhamnolipids production. As shown in Figure 7, Lut significantly reduces the production of all three virulence factors on PAO1 (67 ± 4%; 19 ± 2%; 62 ± 5% inhibition for pyocyanin, elastase and rhamnolipids, respectively) when compared to the DMSO control, suggesting that Lut exerts broad-spectrum inhibition activity towards PAO1 virulence expression.

### 2.6. Lut Disrupts the Expression of QS-Dependent lasB and rhlA Genes in PAO1

As Lut reduces pyocyanin, elastase and rhamnolipids production, which are QS-dependent extracellular virulence factors, we further investigated whether QS-regulated genes could be also affected by Lut. Thus, we assessed the effect of Lut on the expression of *lasB* encoding LasB elastase and *rhlA* encoding the precursor of rhamnolipids. Interestingly, Lut at 22 µM significantly affected *lasB* expression (60 ± 4% of inhibition) compared to the control condition (Figure 8A). A similar effect is observed in *rhlA* expression, which was significantly inhibited (by 62 ± 2%) when grown on LB supplemented with Lut in comparison with the control (Figure 8B). As shown in Figure 8C, Lut has no effect on the transcription of the QS-independent *aceA* gene, suggesting that the effect recorded on QS-regulated gene expression is specific and does not result from a global inhibition of PAO1 metabolic activity.

### 2.7. Lut Affects the Expression of lasR/I and rhlR/I Genes in PAO1

As Lut specifically impacts on QS-dependent *rhlA* and *lasB* expression, we evaluated its impact on the expression of *lasR*/*I* and *rhlR*/*I* that positively control the expression of *rhlA* and *lasB* [13]. Lut at 22 µM significantly reduced the expression of the *lasR* and *lasI* genes (by 60 ± 3% and 56 ± 5%, respectively, Figure 9A) and of the *rhlR* and *rhlI* genes (by 64 ± 3% and 60 ± 3%, respectively, Figure 9B). This suggests that Lut impairs QS at the level of both *las* and *rhl* systems circuitry in PAO1, indicating a plausible synthesis inhibition of the native AHLs or of QS proteins, which consequently reduces QS-regulated genes expression and the production of pyocyanin, elastase and rhamnolipids (Figure 7).

### 2.8. The Exogenous Addition of AHLs (3-oxo-C-12HSL and C4-HSL) Affects the Production of Pyocyanin and Elastase on Both Lut-Treated PAO1 Wild-Type and AHL-Mutant Strains

To address the implication of AHLs in the Lut inhibition process and the matter of whether AHLs supply could restore the production of QS-virulence factors pyocyanin and elastase in the presence of Lut, 3-oxo-C12-HSL (3oxo) or C4-HSL (C4) were added exogenously to Lut-treated PAO1 cells or AHL-mutant strains.

Regarding pyocyanin production, the addition of C4 at 10 μM does not increase the production of pyocyanin in wild-type PAO1 cells in control DMSO condition (Figure 10A). Interestingly, although Lut drastically reduces pyocyanin production (Figure 7A), the addition of exogenous C4 partially restores the production of pyocyanin (+36% of production) in wild-type PAO1 cells (Figure 10A). The same experiment was performed with the ΔPA3476 (∆*rhlI*) mutant strain (which lacks functional *rhlI* synthetase gene and fails to produce pyocyanin) to avoid interference with native C4. As shown in Figure 10B, the ΔPA3476 (∆*rhlI*) mutant, either in the presence of DMSO or of Lut, was unable to produce pyocyanin; however, in both conditions, the addition of exogenous C4 leads to detectable pyocyanin production. However, pyocyanin production in DMSO + C4-induced ∆*rhlI* cells was two-fold higher than that observed in Lut + C4-induced ∆*rhlI* cells suggesting that restoration of pyocyanin production by C4 is partially limited by the presence of Lut.

With regard to elastase production, the addition of 3oxo (10 µM) increases the production of elastase in wild-type PAO1 cells in DMSO condition (+26% of production) (Figure 10C). By contrast, the elastase production, reproducibly inhibited by Lut (Figure 7B), is not restored by the addition of 3oxo at 10 μM (Figure 10C). The same experiment was performed with the ΔPA1432 (∆*lasI*) mutant strain (which lacks functional *lasI* synthetase gene and fails to produce endogenous 3oxo and elastase). As shown in Figure 10D, the ΔPA1432 (∆*lasI*) mutant in the presence of DMSO as well as in the presence of Lut alone was unable to produce elastase, whereas addition of exogenous 3oxo in DMSO but not in Lut conditions leads to detectable elastase production (Figure 10D). This suggests that an exogenous supply of 3oxo is not able to reactivate the elastase production in the presence of Lut.

These results comfort the plausible Lut-induced decrease in native AHLs, particularly for C4 as a partial compensation of pyocyanin production occurs with exogenous C4 addition (Figure 10C). In addition, these data suggest that the decrease in 3oxo, which may occur in the *las* systems, does not entirely explain the origin of the inhibition of elastase production in the presence of Lut.

### 2.9. Lut Affects the Expression of Global Regulator vfr but Not gacA PAO1 Genes

In the hierarchical *P. aeruginosa*-QS cascade, the global regulators GacA and Vfr are positioned upstream to the core QS-machinery and exert a positive effect on the transcriptional regulators LasR and RhlR [17,18,19]. As the inhibition process of Lut seems to operate beyond the simple inhibition of AHLs synthesis in the QS pathways, the effect of Lut on *gacA* and *vfr* expression was investigated. As shown in Figure 11, Lut significantly inhibits the *vfr* expression (16 ± 3% inhibition) but not the *gacA* expression, whereas the positive controls quercetin (QUE) and azithromycin (AZM) inhibit *vfr* and *gacA* expressions, respectively (by 30 ± 3% and 20 ± 4%). These results suggest that Lut either hits multiple targets or primarily impacts *vfr* by inhibiting its expression, thereby inducing a cascade of downstream inhibitions that results in an overall inhibition of the QS machinery, the virulence factors production, the cell motility and the biofilm formation.

## 3. Discussion

The xanthophyll lutein is a carotenoid naturally abundant and available in fruits, cereals, and vegetables with reported antioxidant and anti-inflammatory properties and beneficial effects on eye diseases [35,36]. Interestingly, Songca et al. [37] pointed out the antibacterial potential of Lut extracted from leaves of *Rhus leptodictya* against *P. aeruginosa* with a MIC value of 105 µM. More interestingly, Sampathkumar et al. [30] reported that lutein from the green microalga *Chlorella pyrenoidosa* as well as commercially available Lut both inhibit biofilm formation, pyocyanin and EPS production in PAO1 at 35 µM, indicating a plausible interaction with QS proteins through molecular docking analysis and down-regulation of the *las* and *rhl* genes expression by qRT-PCR analysis.

The present investigation comforts the reported antibiofilm activity of Lut and highlights a reproducibility of inhibition effects in different experimental protocols (different culture media), suggesting that the biofilm inhibition process is probably not influenced by environmental modifications and particularly nutritional parameters. Moreover, we further show that Lut not only affects the phenotype structure of PAO1 biofilm but also a pre-formed biofilm in which the bacterial community is mainly composed of isolated bacterial clumps (Figure 4). In comparison to other reported antibiofilm compounds evaluated through the same experimental procedure, i.e., oleanolic acid, cassipourol and oleanolic aldehyde coumarate (OALC), Lut (22 µM) exhibits a similar inhibition effect (~56%) at a ten- to forty-fold lower concentration. Indeed, biofilm formation is inhibited at ~52%, ~50% and ~44% in the presence of 800 µM oleanolic acid (Figure 2), 200 µM cassipourol and 200 µM OALC, respectively [23,24]. More interestingly, at the phenotypical level, Lut appears more efficient in disrupting biofilm structure, as observed by the high proportion of isolated bacterial clumps in fluorescence microscopy (Figure 3). Indeed, although biofilm disruption is observed in cassipourol- and OALC-treated PAO1 biofilms, a long chain of cell-to-cell connection remains visible, suggesting that bacterial interactions are still preserved in some zones. Nevertheless, all three compounds, in combination with tobramycin, enable the killing of ~90% of one-day-old biofilm-encapsulated PAO1, whereas tobramycin alone could barely kill 20–40% of the bacteria. Finally, it is also interesting to note that Lut still exerts antibiofilm activity at the concentration of 2.75 µM (20 ± 4% inhibition, Figure 2), indicating a strong affinity with its potential target.

Beyond the confirmation that Lut impairs QS-regulated (*lasB* and *rhlA*) and QS-regulator (*lasR*/*I* and *rhlR*/*I*) genes expression in a specific manner (as the QS-independent *aceA* gene was not affected), we highlight that the expression of the *vfr* gene is also inhibited, suggesting that this global regulator of virulence factors is one of the main targets of Lut. Vfr is a cyclic adenosine monophosphate (cAMP)-dependent transcription factor that coordinates the expression of more than 200 genes, including those implicated in the pilus formation machinery (i.e., type IV pili) and QS systems [19]. More precisely, the *vfr* expression is generally activated by increased level of cAMP, by bacterial attachment to solid surface; it is also temporally expressed during the transition from exponential to stationary growth phase [20,38]. Following its binding to cAMP, Vfr regulates its own expression but also the expression of multiple other genes implicated in the virulence of PAO1, including those for Type III secretion system, motility process and QS systems [39]. In the case of motility and QS, Vfr is known to positively regulate (*i*) type IV pili (TFP), surface-exposed fibres that mediate many functions in bacteria, including locomotion (twitching and swarming) and adherence to host cells [40,41], and (*ii*) the *las* system, through direct binding of Vfr to target the promoter of the LasR-encoding gene [17,19,42]. In the same line, Vfr can directly bind the *rhlR* promoter region with positive or negative impacts on *rhlR* transcription; this is modulated through the presence of several Vfr-binding sites in the *rhlR* promoter region [20].

With respect to the important role of QS systems in virulence expression and in the establishment/maturation of biofilms in PAO1, a model in which disruption of Vfr circuitry, leading to an inhibition cascade down to biofilm degradation, is highly plausible especially since *vfr* mutant strain is known to exhibit reduced biofilm formation [43]. Therefore, we hypothesize a model in which Lut inhibits *vfr* expression in PAO1, leading to the reduction in Vfr production, which consequently reduces the expression of Vfr-dependent genes, including the *TFP* genes (particularly *pilY1*), QS-regulator *lasR* (encoding for LasR protein) and *rhlR* (encoding for RhlR protein) (Figure 12). Reduced activity of *TFP* genes leads to reduction in twitching motility, whereas the lack of LasR and RhlR proteins in the *las* and *rhl* systems leads to a reduction in LasR/3-oxo-C12-HSL and RhlR/C4-HSL complexes. These complexes are required for the full activation of QS-dependent genes, including *lasI* (encoding for LasI synthetase), *lasB* (encoding for elastase), *rhlR*, *rhlI* (encoding for RhlI synthetase), *rhlA* (encoding for rhamnolipids precursor, 3-(hydroxyalkanoyloxy) alkanoic acid (HAA)), *phzA1* (encoding for phenazine pyocyanin) and *pel* operon (involved in biosynthesis of a glucose-rich matrix exopolysaccharide) [16,44]. Thus, reduced activation of these QS-regulated genes leads to a drastic decrease in EPS matrix components (particularly eDNA and Pel exopolysaccharide) and virulence factors including elastase, pyocyanin [45] and rhamnolipids with a negative impact on twitching and swarming motilities [46,47]. Finally, these inhibitions lead to the decrease in biofilm formation and maintenance, the reduction in biofilm protective barrier and the dispersal of biofilm.

These three final consequences are largely argued for in the literature. Firstly, the Lut inhibition of biofilm formation and maintenance is associated with disruption in twitching, swarming motilities and rhamnolipids. It is indeed well established that the TFP-mediated twitching motilities play important roles in the attachment onto surfaces and are required in the initiation and maturation of *P. aeruginosa* biofilm [12,48]. Likewise, HAA and rhamnolipids are known to promote twitching/swarming motility and to be involved in forming microcolonies and maintaining open channels that facilitate three-dimensional mushroom-shaped structures formation in *P. aeruginosa* biofilms [46,49]. Secondly, the final process of biofilm lifestyle cycle, i.e., the bacterial dispersal, generally occurs following erosion/sloughing of biofilm matrix and/or initiation of bacterial seeding/migration out of biofilm matrix [50]. Accordingly, the Lut-promoted biofilm dispersal could occur through erosion of biofilm matrix following a reduction in EPS components synthesis, as shown by Sampathkumar et al. [30]. Thirdly, the bactericidal effects of the Lut-tobramycin combination on biofilm-encapsulated PAO1 cells (Figure 5), is most probably due to EPS synthesis reduction that leads to considerably lower protective performances of the biofilm matrix. Indeed, EPS which is a highly hydrated polar mixture of biomolecules, notably comprising exopolysaccharides, extracellular DNA (eDNA) and proteins, contributes not only to the overall architecture of biofilms but also to resistance phenotypes [51]. Similarly, pyocyanin reduction could also be indirectly associated with this biofilm protection failure. In fact, beyond its involvement in host cell degradation and the production of reactive oxygen species inducing oxidative stress [52,53], pyocyanin is also indirectly implicated in the biofilm formation process by promoting the release of eDNA, an important key factor in the establishment and protective properties of *P. aeruginosa* biofilm [54,55].

Lut inhibits *vfr* gene expression in PAO1, as demonstrated in the present study (PS), driving to an inhibition cascade that finally leads to the decrease in biofilm formation and maintenance, reduction in biofilm protective barrier and promotion of biofilm dispersal as schematically symbolized by blue boxes and arrows (data from the present study) and grey boxes and black arrows (literature data: 1, [38]; 2, [20]; 3, [56]; 4, [16]; 5, [44]; 6, [45]; 7, [46]; 8, [47]). The green declining arrows represent genes and metabolites inhibited by Lut, as previously reported by Sampathkumar et al. [30].

To the best of our knowledge, very few natural products have been demonstrated to disrupt Vfr and/or *vfr* expression. The auranofin, a gold-based antirheumatic agent, (20 µM) inhibits QS, Type III secretion system and TFP formation while attenuating biofilm maturation [57]. These effects are in part attributed to cysteines targeting in Vfr that abolish its binding to promoters of *vfr*-mediated genes. Furthermore, auranofin displays synergistic effects in eradicating PAO1 biofilms in vitro and in vivo, when used in combination with the antibiotic colistin. Likewise, the dietary flavonoid quercetin (53 µM) has been reported to inhibit PAO1 biofilm formation, swarming motility and virulence factors, including pyocyanin and protease, in PAO1 [25]. Moreover, as measured by qRT-PCR, this flavonoid inhibits *vfr* expression in PAO1 as well as in *rhlI-, lasI-* and *lasI*/*rhlI*-mutant strains, leading to the conclusion that quercetin probably affects these QS-associated virulence factor via *vfr*, thereby inhibiting PAO1 biofilm formation. Thus, according to the literature, Lut represents the third natural compound that impairs Vfr circuitry and particularly *vfr* expression, leading to profound reduction in PAO1 invasive ability. However, further studies are needed to confirm whether Lut also impedes Vfr at post-translational level (as does auranofin) and whether Vfr is its only target. As quercetin and auranofin, Lut could be a relevant candidate for antivirulence therapy combined with antibiotic treatment; this would be especially interesting as this xanthophyll is already used in dietary supplements without any reported toxicity [35].

Interestingly, these three Vfr inhibitor compounds present very different chemical backbone structures, giving the expectation that many other compounds could exhibit similar inhibition properties. In this regard, it is interesting to note that an oxidative derivative of Lut, 3′-dehydrolutein, exhibits a very similar antivirulence activity (Appendix A) suggesting that in the Lut chemical backbone, the presence of a ketone functional group instead of a hydroxyl group in position C_3_ does not influence antivirulence properties. Moreover, zeaxanthin, an isomer of Lut differing by the position of a double bond, that was selected by in silico screening from a library of 638 lichen metabolites, has been also shown to inhibit biofilm formation and *lasB*/*rhlA* expression in PAO1 at 12 µM [31] suggesting that other Lut isomers may exert similar properties; Lut is also known to generate various *cis*-isomers [58]. All these isomers, but also other xanthophylls and their esters, would be important to test to establish a structure–activity relationship; this would shed light on the chemical functions required for the anti-QS and antibiofilm activities in the perspective to identify active and stable compounds for drug development, with favorable pharmacokinetic properties.

Finally, since Vfr regulates multiple other genes implicated in virulence expression of PAO1, even outside of the QS system pathways, such as the Type III secretion system [39], and with respect to the drastic virulence expression inhibition caused by *vfr* inhibitors, which correlates with *vfr* mutant strain phenotypes, the Vfr circuitry (*vfr* gene and/or its product) appears as a privileged target in screening for PAO1 antivirulence compounds.

## 4. Material and Methods

### 4.1. Bacterial Strains and Growth Conditions

*P. aeruginosa* PAO1 wild-type and its derivatives strains were grown (37 °C, agitation 175 r.p.m) in LB-MOPS broth (50 mM, pH 7.2, Sigma-Aldrich) supplemented with carbenicillin (300 µg/mL) (Appendix A). To determine the QS activity, the used strain and the experimental procedure were as previously described [59,60]. *P. aeruginosa* PAO1 mutant strains were obtained from the Transposon Mutant Collection (Department of Genome Sciences, University of Washington; http://www.gs.washington.edu/labs/manoil/libraryindex.htm accessed on 10 July 2019 and include mutants 11,174 (ΔPA1432, Δ*lasI*) and 32,454 (ΔPA3476, Δ*rhlI*) [61]. When required, the medium was supplemented with 10 μM (final) 3-oxo-C12-HSL or C4-HSL as previously described [27].

### 4.2. Chemicals and Solvents

Solvents of analytical grades were obtained from VWR International (Leuven, Belgium) and redistilled before use. Chemicals of analytical grades were purchased from Sigma Aldrich (St. Louis, MO, USA). *p*-iodonitrotetrazolium chloride (INT), naringenin (NAR), naringin (NIN), oleanolic acid (OA), quercetin (QUE) and antimicrobial drugs (tobramycin and azithromycin) were purchased from Sigma-Aldrich and TCI (Tokyo Chemical Industry Co. LTD, Tokyo, Japan), respectively. Standard all-*trans* lutein (Lut) was purchased from LGC standards (LGC Standards SARL, Molsheim, France). All the drugs tested in this study are diluted in dimethylsulfoxide (DMSO) to the desired concentration.

### 4.3. Assessment of Bacterial Growth Kinetics

The relative growth of *P. aeruginosa* PAO1 at 37 °C with agitation (175 r.p.m) was evaluated by measuring the cell turbidity at A_600_ with a SpectraMax M2 device (Molecular Devices, San Jose, CA, USA) over a 22 h period.

### 4.4. Biofilm Formation and Quantification

The quantification of the biofilm formation was performed on 24-well microplates by using crystal violet staining protocol, as previously described [60]. Briefly, an overnight culture of *P. aeruginosa* PAO1 was washed twice and diluted in fresh biofilm broth (BB) medium (Na_2_HPO_4_, 1.25 g/L; FeSO_4_.7H_2_O, 0.0005 g/L; glucose, 0.05 g/L; (NH_4_)_2_SO_4_, 0.1 g/L; MgSO_4_.7H_2_O, 0.2 g/L and KH_2_PO_4_, 0.5 g/L). The diluted cultures (50 µL) were added to 940 µL of BB medium (initial A_600nm_ of the culture between 0.14 and 0.16) and supplemented with 10 µL of the desired solution. DMSO (1% final concentration) and OA (800 µM, final concentration) were used as negative and positive controls, respectively [62]. Lut was tested at different concentrations (from 1.375 to 88 µM, final concentration). The PAO1 cultures were incubated statically for 24 h at 37 °C and the formed biofilm stained by crystal violet was measured at A_590nm_ with a SpectroMax M2 device (Molecular Devices, San Jose, USA).

### 4.5. Biofilm Phenotype and Synergistic Activity with Tobramycin in Fluorescence Microscopy

The biofilm formation phenotype by *P. aeruginosa* PAO1 cells in glass coverslips cultures was examined in epifluorescence microscopy. In order to evaluate the effects of DMSO (1%, *v*/*v*) or Lut (22 µM) on biofilm development and on one-day-old biofilm, two distinct assays were adopted. The first assay follows the same culture conditions as described in Section 4.5. After 24 h incubation, the biofilm development was visualized using the LIVE/DEAD^®^ BacLight™ bacterial viability kit (Invitrogen, Molecular probes, ThermoFisher Scientific, California, USA). The growth medium was removed and replaced by 500 µL of a solution of SYTO 9 and propidium iodide diluted 400-fold in BB medium. Biofilms were incubated for 15 min and *P. aeruginosa* PAO1 cells were examined under an epifluorescence microscope coupled to a Sony L2 camera using a 40× objective lens and images were false colored and assembled using Microsoft Photos software. For the second assay, PAO1 cells were grown statically in BB medium for 24 h at 37 °C in 24-well polystyrene plates to form biofilms. Tested molecules as described above (DMSO 1%, *v*/*v*; Lut, 22 µM) were added and incubated for a further 24 h, and the biofilm development and bacterial viability in biofilms were assessed as described for the first assay.

The bactericidal activities of tobramycin combined with Lut 22 µM or DMSO 1% in one-day-old biofilm-encapsulated PAO1 cells were also assessed. Tobramycin was chosen because it has been shown that QS inhibition greatly enhances the sensitivity of *P. aeruginosa* to this antibiotic and increases clearance of *P. aeruginosa* in a foreign-body infection model [32]. PAO1 cells were grown statically in BB medium for 24 h at 37 °C in 24-well polystyrene plates to form biofilms. Tested molecules as described above, and tobramycin (50 μg/mL = 107µM) were added and incubated for a further 24 h and the biofilm development and bacterial viability in biofilms were assessed as described for the first assay. The proportion of live or dead bacteria was estimated by counting the observed green and red cells in five different fields. Yellow-colored cells, which probably represent a beginning of loss of membrane integrity, were considered as live bacteria.

### 4.6. Motility Assay

The swarming and twitching motilities were examined by using LB agar plates (0.6, and 1%, respectively) as described previously [60]. At room temperature, before the sterilized and cooled (45–50 °C) LB agar was poured into compartmented Petri dishes, the test solutions were added [DMSO 1%, control condition, Lut 22 µM, final concentration), naringenin at 1 mM or naringin at 1 mM final concentration (positive and negative control for swarming motilities) [60] or azithromycin (positive control for twitching, 2.6 µM) [63]]. Five microliters of bacterial culture (A_600nm_ = 1) were inoculated at the center of each compartment of the Petri dishes and incubated at 37 °C for 24 h (for swarming) or 48 h (for twitching motility). Bacteria spreading from the inoculation spot were measured with sliding caliper. For twitching motility, the agar was discarded from Petri dish; twitching motility zones were visualized by staining for 1 min with 0.1% (*w*/*v*) of crystal violet as proposed by Darzins et al. [64] and diameters were measured.

### 4.7. Quantitative Analysis of Pyocyanin, Elastase and Rhamnolipids Production

Quantification of pyocyanin production was assessed according to previously described procedures [59]. *P. aeruginosa* PAO1 or mutants were grown in LB-MOPS broth for 18 h at 37 °C with agitation (175 r.p.m) and the cells were washed twice in fresh LB medium. Then, 50 µL of PAO1 cell suspension were added to 940 µL of LB-MOPS (starting OD_600nm_ ranged between 0.02 and 0.025, corresponding to 10^7^ CFU/mL) supplemented with 10 µL of Lut dissolved in DMSO (22 µM final concentration) or 10 µL of DMSO (1%, *v*/*v*) or naringenin (4 mM final concentration). After 18 h of growth, samples were taken to assess growth (OD_600nm_) and the remaining volume was used for pyocyanin determination. When required, the medium was supplemented with 10 μM of 3-oxo-C12-HSL or C4-HSL as described previously. Rhamnolipids were extracted and quantified by a methylene-blue-based method, as described by [65]. The elastase production was assessed through the measurement of elastase activity using elastin–Congo red [66].

### 4.8. Gene Expression and Beta-Galactosidase Measurements

The *β*-galactosidase activity induced by reporter genes was measured using o-nitrophenyl-*β*-D-galactopyranoside to monitor gene expression. Briefly, PAO1 reporter strains were washed twice in fresh LB medium, resuspended in liquid LB-MOPS-carbenicillin, and grown at 37 °C and 175 r.p.m. for 18 h. PAO1 reporter strains were washed twice in fresh LB medium and resuspended in liquid LB-MOPS-carbenicillin. PAO1 reporter strain inoculum (50 µL) was incubated (37 °C with agitation at 175 r.p.m.) for 18 h in 940 µL LB-MOPS-carbenicillin (initial OD_600nm_ of the culture was between 0.020 and 0.025) supplemented with 10 µL of Lut dissolved in DMSO (22 µM final concentration) or 10 µL DMSO (1%, *v*/*v*). Additionally, naringenin, known QS inhibitor in *P. aeruginosa* [27] was used as positive control. Likewise, azithromycin and quercetin were used as positive controls for *gacA* and *vfr* experiment, respectively [25,67]. After incubation, the bacterial density was assessed by spectrophotometry (OD_600nm_) and the gene expression by the *β*-galactosidase assay.

### 4.9. Statistics

All experiments were performed at least in four replicates and repeated in three independent assays. The data significance (*p*-value ≤ 0.01) was statistically analyzed by conducting Student’s *t*-tests or one-way ANOVA with post hoc Tukey tests using Graphpad Prism 8 software.

## 5. Conclusions

Lut and one of its major oxidative derivatives both inhibit PAO1 virulence expression and biofilm formation via the *vfr*-mediated *las* and *rhl* systems. These findings highlight that Lut structure-based compounds may represent relevant bioactive compounds to limit the virulence expression and invasive ability of *P. aeruginosa* and open the debate for the potential use of Lut beyond its actual application as dietary supplement. To the best of our knowledge, Lut-type carotenoids are probably atoxic, even way over dietary dosages, and have not been qualified as pan-assay interference compounds (PAINS) that could raise suspicion of false positives activities [68]; as such, these carotenoids warrant the attraction of researchers for their integration as antivirulence drug candidates.

Further experiments remain to be carried out to evaluate their effectiveness using in vivo models. In this perspective, a *C. elegans* model would be useful to verify the expected virulence attenuation of PAO1 by measuring an increase in living in *C. elegans* infected by Lut-treated PAO1 or in PAO1-infected *C. elegans* subsequently treated with Lut. Likewise, the use of mouse models with pulmonary or peritoneal implant infection will be informative about the Lut concentration required for a rapid clearance of PAO1. The low effective concentration of Lut observed in vitro gives the expectation that the in vivo effective dose would not exceed an eventual toxic dose threshold.

As other bacterial species are also responsible for persistent bacterial infection, the impact of Lut towards other bacterial strains would be of interest. This mainly concerns bacteria that share similar virulence regulation mechanism with *P. aeruginosa*, i.e., Gram-negative bacteria, and particularly *Escherichia coli*, as this species exhibits high potential pathogenicity and relies on the cAMP-receptor protein, a global regulator of stress resistance and virulence that shares 91% structure similarity with Vfr [69].

Finally, in our quest to identify eligible antivirulence compounds for drug development, the present study gives a glimpse of hope for identifying other compounds active at concentrations as low as some conventional antibiotics (e.g., azithromycin, 2.6 µM) [24].

## Figures and Tables

**Figure 1 ijms-23-07199-f001:**
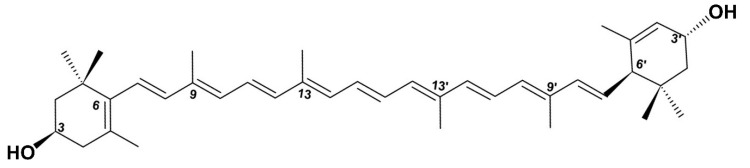
Chemical structure of dietary xanthophyll carotenoid, all-*trans* lutein assigned as (3R,3R′,6′R)-lutein.

**Figure 2 ijms-23-07199-f002:**
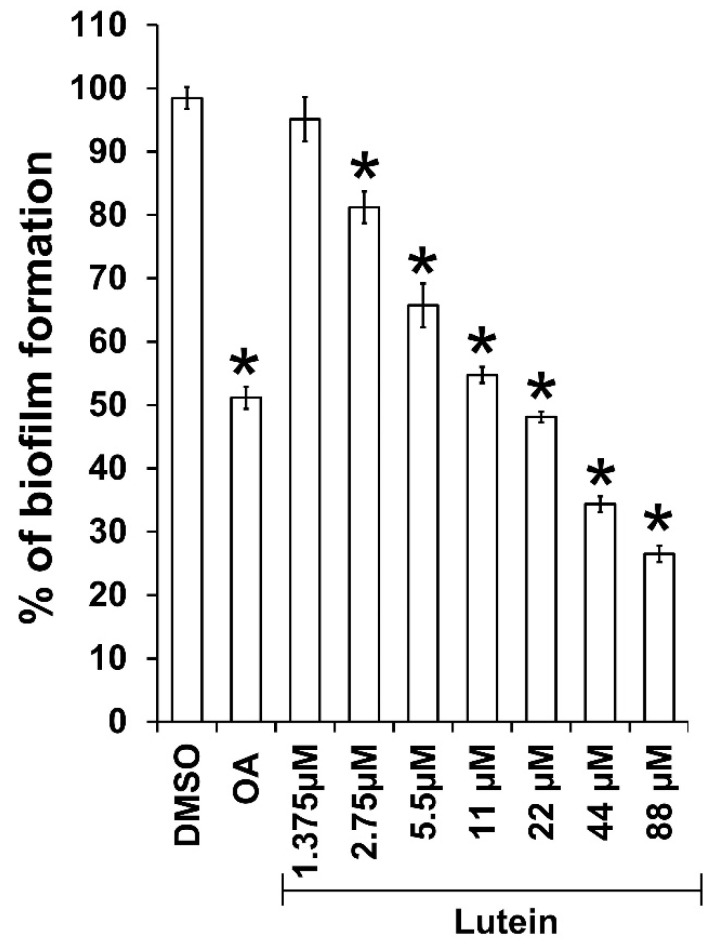
Dose-dependent anti-biofilm activity of Lut. The biofilm formation of *P. aeruginosa* PAO1 grown in minimal medium supplemented with DMSO 1%, oleanolic acid 800 µM (OA, as positive control) or different concentrations of Lut (from 1.375 to 88 µM) after incubation without agitation at 37 °C for 24 h. Biofilm formation was quantified by crystal violet staining and measured as A_590nm_ and expressed as % of biofilm formation. All experiments were performed in quintuplicate with three independent assays. Error bars represent the SEM; asterisks indicate samples that are significantly different from the DMSO (*p* ≤ 0.01).

**Figure 3 ijms-23-07199-f003:**
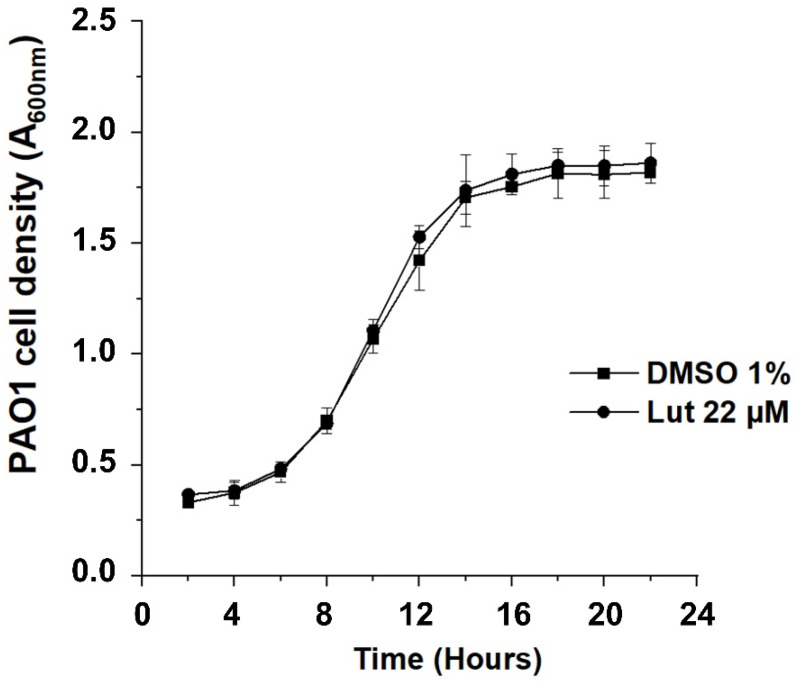
Effect of Lut on the growth kinetics of PAO1. Growth kinetics of PAO1 in presence of Lut at 22 µM or DMSO 1% over a period of 22 h under agitation by measuring the cell density of the bacterial growth at A600 nm. The bars represent the standard error of the mean. There were no statistical differences between the 2 conditions at any time point (n = 4) (*p*-values > 0.01).

**Figure 4 ijms-23-07199-f004:**
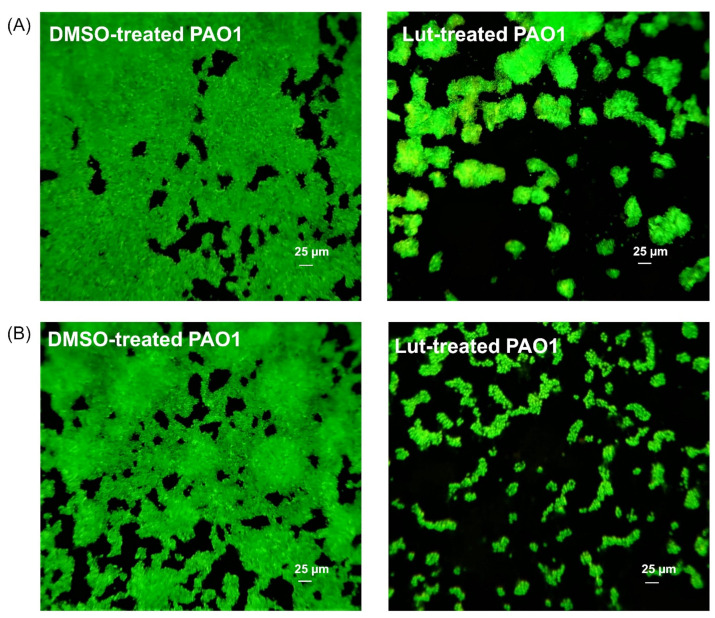
PAO1 biofilm phenotypes as affected by Lut. (**A**) Effect of Lut on biofilm development in which PAO1 cells were incubated statically at 37 °C for 24 h in presence of DMSO 1%, or Lut at 22 µM. (**B**) Effect of Lut on one-day-old preformed biofilm in which PAO1 cells were incubated for 24 h and then treated for 24 h with DMSO 1%, or Lut at 22 µM. Cells were visualized after staining with SYTO-9 (LIVE/DEAD BacLight kit) in which green fluorescence color represents living cells. Fluorescence microscopy was achieved by using an epifluorescence microscope coupled to Sony L2 camera using a 40× objective lens, and images were false colored using Microsoft Photos software.

**Figure 5 ijms-23-07199-f005:**
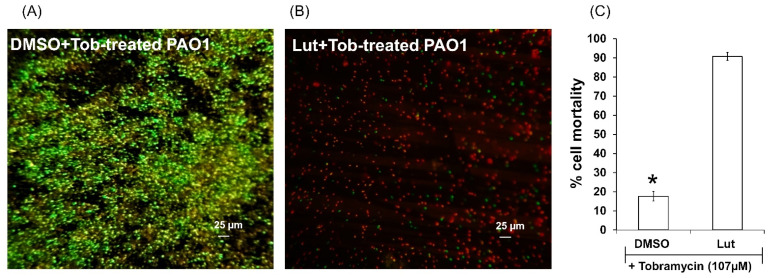
Synergistic activity of Lut and tobramycin against biofilm-encapsulated PAO1. PAO1 cells were incubated statically for 24 h and then treated for 24 h with tobramycin (107 µM final concentration) and DMSO 1% or Lut (22 µM final concentration). (**A**) DMSO + tobramycin, (**B**) Lut + tobramycin. (**C**) Quantification of bacterial viability. Assessment of bacterial viability and microscopy were performed as in Figure 4. Red color indicates dead cells and green/yellow colors indicate alive cells. Error bars represent the standard errors of the means; all experiments were performed in quintuplicate with three independent assays, and asterisk indicates significant difference from the DMSO (*p* < 0.01).

**Figure 6 ijms-23-07199-f006:**
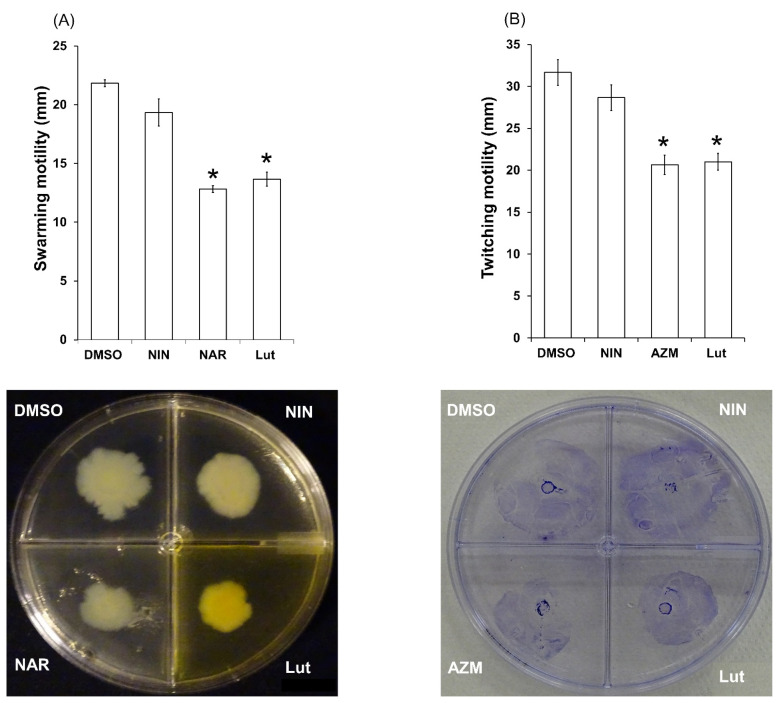
Effect of Lut on *P. aeruginosa* PAO1 motilities (**A**) swarming ability of PAO1 onto LB agar (0.6%) supplemented with glutamate (0.05%) and glucose (0.2%) as well as DMSO (1%), Lut (22 µM final concentration), naringenin (NAR, 1 mM final concentration) or naringin (NIN, 1 mM final concentration). After incubation at 37 °C for 24 h, the zones of migration from the point of inoculation were measured for each condition and expressed in mm. (**B**) Twitching motility of PAO1 onto LB agar (1%) alone (LB) or supplemented with DMSO (1%), or Lut (22 µM final concentration), or azithromycin (AZM, 2.6 µM final concentration). The twitching zones were stained, and their diameters were measured and expressed in mm after incubation at 37 °C for 48 h. All experiments were performed in triplicate with three independent assays. Error bars represent the standard error of the mean and the asterisks indicate samples that are significantly different from control condition (*p* ≤ 0.01).

**Figure 7 ijms-23-07199-f007:**
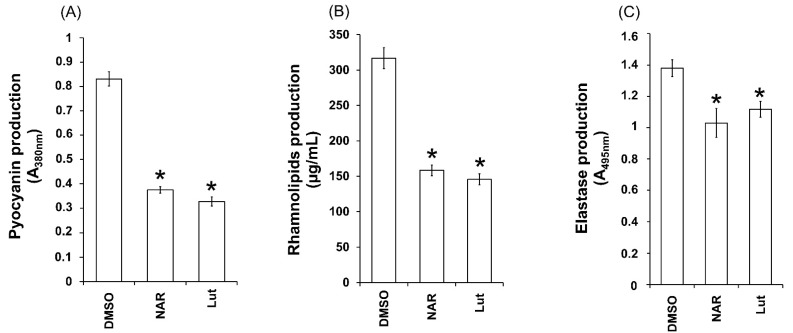
Lut inhibits the PAO1 production of virulence factors: (**A**) Effects of Lut on pyocyanin production; (**B**) Effect of Lut on rhamnolipids production; (**C**) Effect of Lut on elastase production. Lut was tested at 22 µM; naringenin (NAR; 4 mM) and dimethylsulfoxide (DMSO, 1%) are used as positive and solvent control, respectively [27]. Pyocyanin was extracted and quantified by absorbance measurement at 380 nm. The rhamnolipids production was measured using methylene-blue-based method and expressed in µg/mL. Elastase production was assessed by an elastolysis assay and calculated as the ratio between A_495_ and A_600_. All experiments were performed in triplicate with three independent assays. Error bars represent the standard error of the mean and the asterisks indicate samples that are significantly different from control condition (*p* ≤ 0.01).

**Figure 8 ijms-23-07199-f008:**
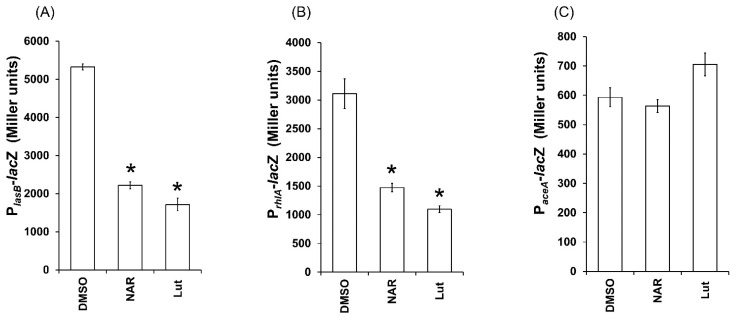
Effect of Lut on the expression of PAO1 QS-regulated *lasB* and *rhlA* and the QS-independent *aceA* after 18 h of incubation. (**A**) Effect of Lut at 22 µM on *lasB* expression. (**B**) Effect of Lut on *rhlA* expression. (**C**) Effect of Lut on *aceA* gene expression. Gene expression was measured as the *β*-galactosidase activity of the *lacZ* gene fusions and expressed in Miller units. Naringenin (NAR) was used as a QS inhibitor control and DMSO as a negative control. Error bars represent SEM, and all experiments were performed in quintuplicate with three independent assays; asterisks indicate samples that were significantly different from the DMSO (*p* ≤ 0.01).

**Figure 9 ijms-23-07199-f009:**
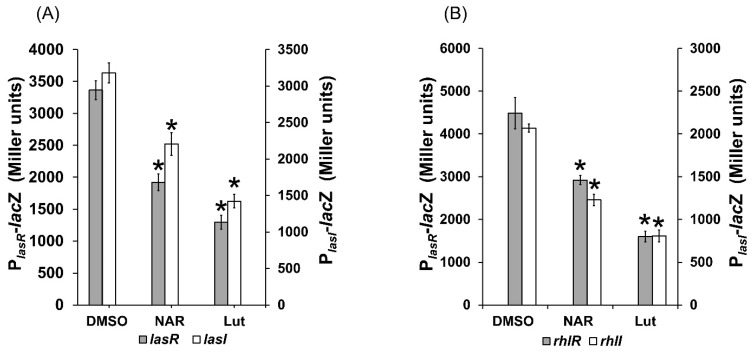
Effect of Lut at 22 µM on the expression of PAO1 QS-regulator genes *lasR*/*I* and *rhlR*/*I* after 18 h of incubation. (**A**) Effect of Lut on *lasR* (grey bar) and *lasI* (clear bar) expression. (**B**) Effect of Lut on *rhlR* (grey bar) and *rhlI* (clear bar) expression. Gene expression was measured as the *β*-galactosidase activity of the *lacZ* gene fusions and expressed in Miller units. Naringenin (NAR) was used as a QS inhibitor control and DMSO as a negative control. Error bars represent SEM, and all experiments were performed in quintuplicate with three independent assays; asterisks indicate samples that were significantly different from the DMSO (*p* ≤ 0.01).

**Figure 10 ijms-23-07199-f010:**
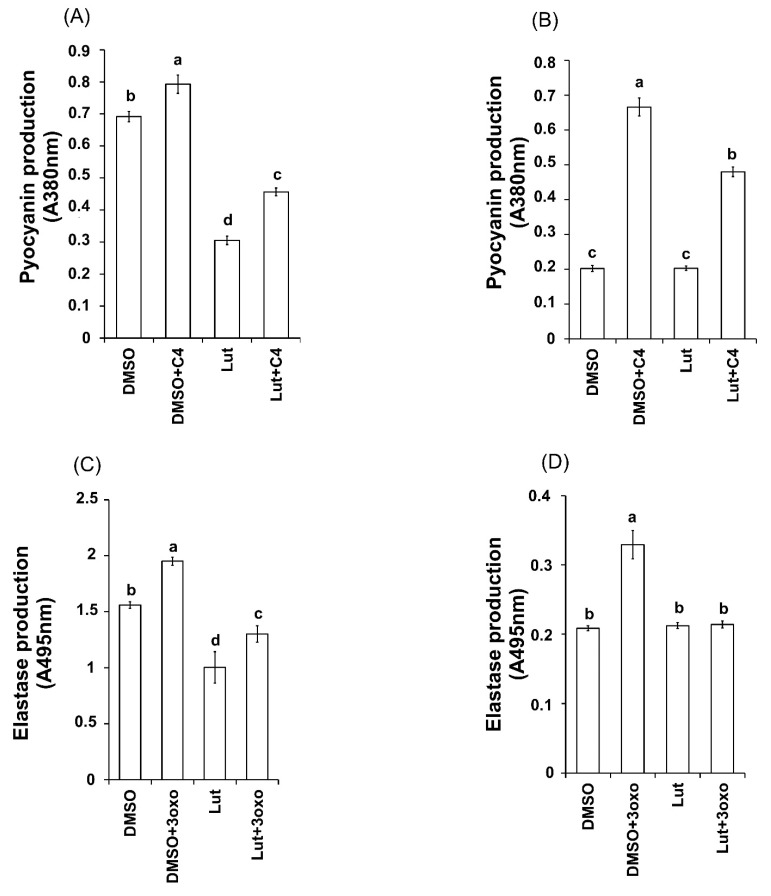
Effect of Lut on the production of pyocyanin and elastase after exogenous supply of AHLs in PAO1 wild-type and mutant strains. (**A**–**D**) Production of pyocyanin and elastase by the wild-type strain PAO1 and ∆*rhII* (∆PA3476), ∆*lasI* (∆PA1432) mutant strains. Pyocyanin was extracted and quantified by absorbance measurement at 380 nm. Elastase production was assessed via an elastolysis assay and calculated as the ratio between A_495_ and A_600_. In each case, bacteria were incubated with DMSO, Lut or the appropriate AHL. Bacteria were also induced with the appropriate AHL and simultaneously treated with DMSO or Lut. C4-HSL (C4) and 3-oxo-C12-HSL (3oxo) were added at 10 µM final concentration. DMSO-treated cultures were used as controls, the statistical significance of each test (n = 4) was evaluated by conducting one-way ANOVA with Tukey’s multiple comparison tests, and a *p*-value of 0.01 was considered significant. Different letters (a, b, c and d) above the bars indicate data that are statistically different from each other (*p* < 0.01).

**Figure 11 ijms-23-07199-f011:**
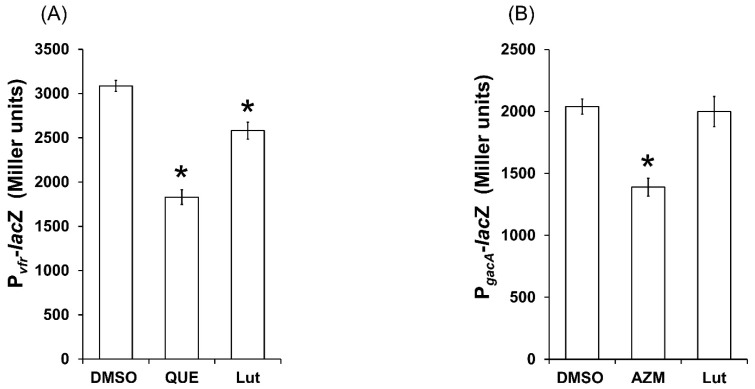
Effect of Lut at 22 µM on the expression of PAO1 global regulator *vfr* and global activator *gacA* genes after 18 h of incubation. (**A**) *vfr* expression in presence of Lut. (**B**) *gacA* expression in presence of Lut. Gene expression was measured as the *β*-galactosidase activity of the *lacZ* gene fusions and expressed in Miller units. Quercetin (QUE) and azithromycin (AZM) were used as *vfr* and *gacA* inhibitor control and DMSO as a negative control. Error bars represent SEM and all experiments were performed in quintuplicate with three independent assays; asterisks indicate samples that were significantly different from the DMSO (*p* ≤ 0.01).

**Figure 12 ijms-23-07199-f012:**
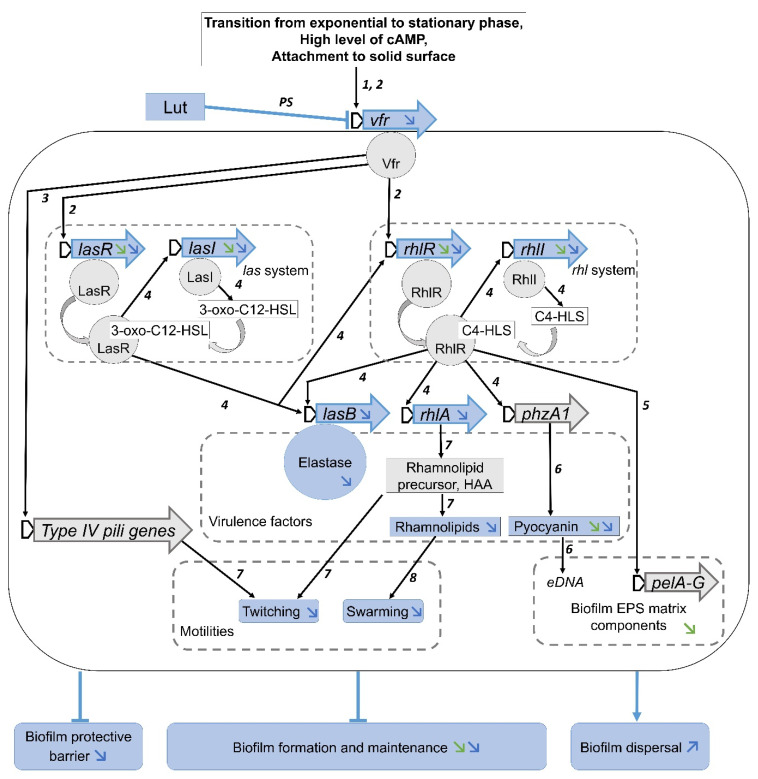
Proposed model of the mechanism underlying the target and inhibition cascade occurring in Lut-treated PAO1 to impact the Vfr transcriptional regulator, the regulation of QS-related genes and motility genes, and the biofilm development, maintenance and protective properties.

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
