# Peer review of "The Xanthophyll Carotenoid Lutein Reduces the Invasive Potential of Pseudomonas aeruginosa and Increases Its Susceptibility to Tobramycin"

_ijms, 2022, doi:10.3390/ijms23137199_

Round 1

Reviewer 1 Report

This is an interesting manuscript describing investigation of lutein's ability to disrupt biofilm formation and to increase susceptibility pf Pseudomonas aeruginosa. to tobramycin. A reasonably well discussed investigation of the mechanism is included. In addition it is demonstrated that an oxidative metabolite of Lut, 3’-dehydrolutein has similar activity. Biofilm formation is indeed an important problem as is antibiotic resistance and thus this paper should be of interest and importance to those involved in anti-infective research. The results here could lead to development of novel treatments for the above. The manuscript is well written and deserves publication after editorial review for typos.

Author Response

Dear reviewer

Thank you very much for giving of your precious time to evaluate our work. Your interest to our work give us more motivation to go on in this particular research field.

Best regards,

Reviewer 2 Report

Dear authors,

Manuscript ijms-1780698entiteled "The xanthophyll carotenoid lutein reduces the invasive potential of Pseudomonas aeruginosa and increases its susceptibility to tobramyci" and authored by Christian Emmanuel Mahavy , Adeline Mol , Blandine Andrianarisoa , Pierre Duez , Mondher El Jaziri , Marie Baucher , Tsiry Rasamiravaka targets a hot topic of considerable interest to the community and journal readers. I really appreciated reading this manuscript. Experiments are accurately designed and nicely conducted. Unfportunately to meet the journal standards some aspects needs author's attention and need to be adressed:

1. Introduction section: please enrich the introduction with mechanisms of natural and synthetic compounds interfering with bacterial virulence, biofilm formation, ...

2. Please discuss the efficiency of lutein in inhibiting biofilm formation compared to oleanolic acid and other interfering compounds with biofilm formation. Please discuss the concentrations used and observed effects.

3. It is really disappointing that conclusion is not well developed given the findings in the paper. Please discuss more widely the impact of your study in the field, the experiments that need to be conducted beyond effectiveness in in vivo models and mainly the possibility to transfer this finding to closely related species.

I am really looking forward receiving an improved version of the manuscript that adresses the issues I highlighted and that meets the journal standards and that I could recommend for publication.

Best regards

Round 2

Reviewer 2 Report

Dear Authors,

Thanks for addressing my comments. I can now recommand your manuscript for publication.

Best regrds